# Ultra-High-Performance Liquid Chromatography Quadrupole Time-of-Flight Mass Spectrometry for Simultaneous Pesticide Analysis and Method Validation in Sweet Pepper

**DOI:** 10.3390/molecules28145589

**Published:** 2023-07-22

**Authors:** Han Yeol Bang, Yong-Kyoung Kim, Hyoyoung Kim, Eun Joo Baek, Taewoong Na, Kyu Sang Sim, Ho Jin Kim

**Affiliations:** 1Gyeongnam Provincial Office, National Agricultural Products Quality Management Service, Busan 47537, Republic of Korea; bbangkkc@korea.kr; 2Experiment Research Institute, National Agricultural Products Quality Management Service, Gimcheon-si 39660, Republic of Korea; ykkim79@korea.kr (Y.-K.K.); hyo02@korea.kr (H.K.); qrgh1004@naver.com (E.J.B.); naratw@korea.kr (T.N.); sim9612@naver.com (K.S.S.)

**Keywords:** measurement uncertainty, method validation, pesticide residue, sweet pepper, UHPLC-QTOF-MS

## Abstract

Pesticides effectively reduce the population of various pests that harm crops and increase productivity, but leave residues that adversely affect health and the environment. Here, a simultaneous multicomponent analysis method based on ultra-high-performance liquid chromatography quadrupole time-of-flight mass spectrometry (UHPLC-QTOF-MS) pretreated by the QuEChERS method was developed to control the maximum residual levels. Among the 140 pesticides with high frequency of detection in agricultural products in Gyeongnam region in Korea for 5 years, 12 pesticides with high detection frequency in sweet pepper were selected. The analytical method is validated, linearities are r^2^ > 0.999, limit of detection (LOD) ranges from 1.4 to 3.2 µg/kg, and limit of quantification (LOQ) ranges from 4.1 to 9.7 µg/kg, and the recovery rate was 81.7–99.7%. In addition, it was confirmed that a meaningful value of these parameters can be achieved by determining the measurement uncertainty. The results proved that parameters such as recovery rate and relative standard deviation of the analysis method were within international standards. Using the developed method, better and safer sweet peppers will be provided to consumers, and effective pesticide residue management will be possible by expanding to other agricultural products.

## 1. Introduction

Pesticides help maintain production by efficiently reducing the population of various pests that harm crops. However, their use also leads to the formation of pesticide residues on crops, which adversely affect health and the environment. Therefore, it is essential to regulate their usage, for which various standards such as CODEX and EU have been developed to manage their maximum residue limits [1]. Pesticides are generally spread in the environment through agricultural water or rainfall, and when highly volatile, disperse as aerosols [2], causing a variety of environmental problems. Therefore, continuous and intensive use of pesticides pollutes the soil and reduces the diversity of plants and animals, thereby threatening the stability of the entire ecosystem [3]. In addition, humans exposed to pesticides can develop various life-threatening diseases such as cancer and genetic disorders [4]. Especially, oral intake is more dangerous than exposure through the skin [5].

Fresh fruits and vegetables are rich in antioxidants such as vitamins and polyphenols [6]. These antioxidants reduce the amounts of free radicals present in the body and prevent damage to DNA and cells of the human body [7]. Sweet pepper contains large amounts of polyphenols, flavonoids, aglycones, and glycosides [8], and these phytochemicals can prevent cardiovascular disease, Alzheimer’s disease, and Parkinson’s disease [9]. Therefore, approximately 3000 ton of sweet pepper is produced and consumed annually worldwide. Sweet peppers are consumed not only for their taste but also for their protective action against various diseases [10].

However, the increase in the production and consumption of sweet pepper has also increased the use of pesticides. Pesticide residues within fruits and vegetables in high concentrations is a major route of pesticide exposure [11], and it is also possible that the pesticide contents in fruits and vegetables may increase or transform into more toxic metabolites during the manufacturing process [12]. Some pesticides remain in the sweet pepper in large amounts even after cooking [13], affecting the health of consumers. Hence, it is essential to develop appropriate preprocessing and analysis methods to determine the presence of pesticide residues in sweet pepper.

Some phytochemicals such as flavonoids and polyphenols interfere with the detection of target analytes through matrix effects [14]. To overcome these effects, suitable preprocessing methods such as solid-phase extraction (SPE) and liquid–liquid extraction (LLE) [15] and QuEChERS for positive matrix effects [16] have been previously employed. Multiple studies have analyzed pesticide residues in sweet pepper by using QuEChERS along with gas chromatography triple quadrupole mass spectrometry (GC–MS/MS) and liquid chromatography (LC) MS/MS [17,18,19]. Nevertheless, unlike the analysis using MS/MS, very few studies have investigated the analysis of pesticide residues in sweet pepper using quadrupole time-of-flight mass spectrometry (QTOF-MS).

QTOF-MS is known to reduce the deterioration of the peak shape that occurs when multiple compounds are screened. In addition, QTOF-MS has a relatively high resolving power that helps minimize the false-positive phenomenon that occurs when similar elements are analyzed [20]. While only a limited number of analytes can be simultaneously investigated through MS/MS, QTOF-MS has a relatively broad spectrum, high sensitivity, and allows for retrospective analysis. Therefore, QTOF-MS is gaining increasing attention as a highly useful tool [21,22]. It has already been used in various fields such as for the analysis of veterinary drug and pesticide residues in pig muscle [23] and phenolic compounds present in plums [24].

In this study, QuEChERS was used as a preprocessing method for pesticide residue analysis in sweet pepper, and UHPLC-QTOF-MS was used to obtain more reliable quantitative and qualitative results than previously developed methods using LC–MS/MS and GC–MS/MS. Method validation was performed using different parameters including measurement uncertainty, and the significance of the experiment was demonstrated by analyzing error factors that may occur during the experiment.

## 2. Results and Discussion

### 2.1. Simultaneous Multicomponent Analyses

From 2015 to 2020, 140 pesticides with a history of detection in the Gyeongnam region of Korea were selected through UHPLC-QTOF-MS analysis, and multiple reaction monitoring (MRM) for pesticides is shown in Appendix A. The precursor ions of all analytes were successfully analyzed within the range of 163.05–746.48 *m*/*z*. Fragment ions of the five analytes with the highest sensitivity were selected. Among these five, the top two fragment ions with the highest sensitivity were selected as representative ions, which were then screened to determine a suitable retention time.

Thereafter, the 12 pesticide residues were selected and used to perform optimization (Table 1). In the case of acequinocyl, cyflumetofen, and procymidone, the experimental *m*/*z* is different from the calculated *m*/*z*, because of the presence of an NH_4_ adduct that increases their stability. No significant difference was observed in the retention time from that reported in previous MRM settings. The most sensitive fragment of each material was selected and used for validation.

### 2.2. Method Validation

Method validation was performed following the guidelines of the European Commission [25] and EURACHEM guides [26]. The method was validated for selectivity, linearity, sensitivity, accuracy, and precision to confirm its effectiveness for the analysis of pesticide residues in sweet pepper (Table 2).

Selectivity was determined based on the presence or absence of interfering peaks in the chromatography. As shown in Figure 1, the selected 12 pesticide residues were observed by conducting separate chromatography analyses within 15 min. The selectivity was found to be excellent and the separation was successful.

Linearity was evaluated based on a calibration curve using five different concentrations (5, 10, 25, 50, and 100 μg/L) of the mixture of each standard. The mixed solution was injected three times for evaluation and a formula was derived using the obtained values. The selected materials showed positive results (r^2^ > 0.999), possibly owing to the high selectivity of QTOF-MS. Hence, it can be suggested that matrix-matched external calibration using a standard can be used for quantitative purposes.

Sensitivity was evaluated by determining the limits of detection (LOD) and limits of quantification (LOQ). The LOD and LOQ were calculated using the standard deviation of the value obtained from multiple replicates of a sample with the lowest concentration (10 µg/kg). Acequinocyl showed the highest LOD, while spirotetramat-enol showed the lowest LOD. The LOQ also exhibited the same trend. The LOD of the 12 pesticide residues ranged from 1.4 to 3.2 µg/kg, while their LOQ ranged from 4.1 to 9.7 µg/kg. Therefore, all 12 residual pesticides used in the analysis were found to have suitable sensitivity for analyzing sweet pepper.

The precision of the experimental method was determined by determining their intraday precisions. The pesticide residue solution was measured three times a day, and the result was expressed as the percentage of the coefficient of variation (CV). The recovery rate for each residual pesticide was calculated by comparing the samples (10, 50, and 100 µg/kg) spiked with sweet pepper blank and standard mixture at each concentration. The following CV values were obtained for the pesticides at different concentrations: 5.4–19.1% for 10 µg/kg, 1.8–7.7% for 50 µg/kg, and 0.8–6.4% for 100 µg/kg. The recovery rate showed a slight difference depending on the concentration of each material, although all the recovery rates were between 80% and 110%. Hence, the proposed experimental method meets the criteria of the presented guideline [25].

Therefore, this method shows an appropriate level of LOD and LOQ, a CV of less than 20%, and a recovery rate of 70 to 120%, as specified in the EU guidelines [25], so that significant results can be obtained in the simultaneous multicomponent analysis of pesticide residues.

Other papers obtained analysis results only using a triple quadrupole, but this paper can more accurately identify the detected pesticide by quantifying it using a triple quadrupole and qualitatively confirming the molecular weight of the pesticide component to four decimal places using QTOF-MS.

### 2.3. Measurement Uncertainty

Measurement uncertainty was calculated according to the Guide to the Expression of Uncertainty in Measurement [27]. The measurement uncertainty is used as an indicator of the reliability of the analysis result by presenting a range estimated to be the actual value. In this experiment, the following uncertainty factors were considered during the analysis: sample weight, final volume, a stock standard used when preparing the calibration curve, and working standard. Various factors such as certification, temperature, and repeatability were also used to estimate uncertainty (Figure 2). First, sample weights and final volumes are common to all analyses. Balance, repeatability, and stability are generally used as uncertainty factors for sample weight, and 10 mL pipettes as uncertainty factors for the final volume (Table 3). The uncertainty in the calibration curve concentration arises owing to the following three factors: stock standard solution (100 mg/L), working standard solution (1 mg/L), and calibration. Standard material purity, balance, and volumetric flask are also considered uncertainty factors as stock standard solutions are prepared at 100 mg/L of the analyte. The working standard solution is a manufacturing process for diluting the stock solution to 1 mg/L, and the stock solution, pipette, and volumetric flask are considered the uncertainty factors. A calibration curve of 5–100 μg/L is prepared by appropriately diluting the working standard solution, and the result of the 10 μg/L addition test is used as an uncertainty factor (Table 4). The relative standard uncertainty is obtained by combining each standard uncertainty, and the relative combined standard uncertainty, combined standard uncertainty, and expanded uncertainty are sequentially obtained, and finally the measurement uncertainty is calculated. The result of the calculated measurement uncertainty was between 9.1% and 18.6% (Table 5). Spirodiclofen showed the highest measurement uncertainty of 18.6%, although the guidelines on measurement uncertainty suggested that a value of <44% was significant at a measurement concentration of 10 µg/kg [28]. Therefore, it can be concluded that all compounds meet the criteria for measurement uncertainty. By calculating the measurement uncertainty for 12 pesticides, errors that may occur during the experiment were confirmed and minimized.

### 2.4. Application of the Developed Method to Sweet Peppers

A total of 276 sweet pepper samples were collected from 15 areas in the Gyeongnam region in Korea. The number of samples of each city are different; details of the sample number and collecting area are shown in Table 6. The results show that 12 pesticides were analyzed for all samples and 10 pesticides were detected in 234 samples. Two pesticides, spirodiclofen and spirotetramat-enol, were not detected in the analyzed sweet pepper, and procymidone was detected in only two samples from Jinju. The total number of detections was 101 for boscalid, 81 for flonicamid, and 60 for pyridaben and spirotetramat. Boscalid, flonicamid, pyridaben, and spirotetramat, which have high detection frequencies, were detected at concentrations of 0.011–1.316, 0.01–0.485, 0.01–0.964, and 0.016–1.626 mg/kg, respectively (Table 6 and Appendix A). The concentration of total pesticides detected ranged from 0.01 to 1.626 mg/kg. The Haman samples showed 43.2 and 28.7% detection rates of flonicamid and boscalid, respectively (Appendix A). The obtained analysis data were quantified using the quadrupole mode, and qualitatively confirmed using the QTOF mode to confirm the results. All 10 pesticides detected in sweet pepper are considered to be safely managed below the maximum residue limits (MRLs) in Korea, which are regulatory limits that set the level of pesticides allowed to remain in foods to protect human health.

## 3. Materials and Methods

### 3.1. Chemicals and Reagents

Water, acetonitrile, and methanol was purchased from Merck KGaA (Darmstadt, Germany) and used as the solvents in the overall experiment with steps such as extraction and dilution of the sample. Formic acid (98%) and ammonium acetate (99%), required for solvent composition, were purchased from Sigma-Aldrich (Steinheim, Germany). The pesticide residue standards were purchased from Agilent Technologies (Santa Clara, CA, USA). The QuEChERS extraction kit and 2 mL of QuEChERS dispersive SPE used for purification were obtained from Agilent (Boblingen, Germany). The sweet pepper was purchased from a market in Gyeongnam and kept refrigerated at −4 °C.

### 3.2. Instrumentation and Conditions

Analysis of pesticide residues using UHPLC-QTOF-MS was performed as follows. First, a 5200 NASCA2 HPLC (Osaka, Japan) with a Waters ACQUITY UPLC BEH C18 (2.1 mm × 100 mm, 2.7 μm) column was used. The composition of mobile phases and other conditions such as gradient compositions and ion mode are listed in Table 7. Mass spectrometry detection was performed using QTOF (AB Sciex X500R QTOF, Sciex, Framingham, MA, USA), and final data processing was performed with SCIEX OS software (version no. 1.7.0.36606).

### 3.3. Sample Preparation

The sample was homogenized using a grinder (T 25 digital ULTRA-TURRAX^®^, IKA, Staufen, Germany). After weighing 10 g of the sample, 10 mL of acetonitrile was added to each weighed sample and shaken for 1 min. Thereafter, a QuEChERS extraction kit (magnesium sulfate: 98.5–101.5%; sodium chloride: ≥99.5%; sodium citrate: 99.9%; disodium citrate sesquihydrate: 99%) was added to the sample solution, followed by vigorous shaking for 1 min using a rotary mixer (DE/VIVA, Collomix GmbH, Gaimersheim, Germany). Subsequently, centrifugation was performed for 10 min at 4000 rpm using SORVALL LYNX 4000 (Thermo Scientific, Waltham, MA, USA). Then, 1 mL of the supernatant was put into the QuEChERS dispersive SPE kit (primary secondary amine, octadecyl silane end-capped, magnesium sulfate; 98.5–101.5%), mixed with Mixmate 5353 (Effendorf, Hamburg, Germany) for 1 min, and centrifuged again with Minispin plus 545 (Effendorf, Hamburg, Germany) at 10,000 rpm for 1 min. The liquid separated through this process was filtered with a 0.2 μm PTFE syringe filter (Whatman, Maidstone, UK) and was used as the final sample.

### 3.4. Standard Sample Preparation and Method Validation

Pesticide residues used as standards were prepared at a concentration between 1000 and 2000 mg/L, diluted with acetonitrile, and mixed to set the appropriate concentration (5, 10, 25, 50, and 100 µg/L). Next, the working standard was mixed with a blank extract to obtain a matrix-matched standard. Multiple simultaneous analysis conditions were established using the standard. Afterward, based on the monitoring results obtained under the set conditions according to Section 2.2 and Section 2.3, the 12 most detected pesticide residues were selected as the main compounds for validating the method. Their calibration curves were prepared by matching them with those of the matrix working solutions. Working solutions were mixed with the sweet pepper extract to produce a matrix-matched sample, which was used as the final sample to determine the parameters (selectivity, precision, accuracy, sensitivity, and linearity) for method validation and to measure the uncertainty values of the experiment. The matrix effect (ME) was calculated by the following equation [29]:$$ME(\%)=(\frac{\mathrm{Slope}\mathrm{of}\mathrm{clibration}\mathrm{curve}\mathrm{in}\mathrm{matrix}}{\mathrm{Slope}\mathrm{of}\mathrm{clibration}\mathrm{curve}\mathrm{in}\mathrm{solvent}}-1)\times 100$$

## 4. Conclusions

Qualitative and quantitative results were obtained through a simultaneous analysis method using UHPLC-QTOF-MS for 12 residual pesticides found in sweet pepper. The analyte was quickly extracted through acetonitrile-based QuEChERS pretreatment, and the method was verified through various parameters such as selectivity, linearity, sensitivity, accuracy, and precision. As a result, all parameters conformed to international standards, proving the validity of the experimental method. In addition, the reliability of the measurement result was calculated as a quantitative indicator by calculating the measurement uncertainty, thereby proving that the experimental result was meaningful. When actual sweet peppers were analyzed using this verified method, 10 pesticides out of 12 were detected and all were detected below the MRLs in Korea.

## Figures and Tables

**Figure 1 molecules-28-05589-f001:**
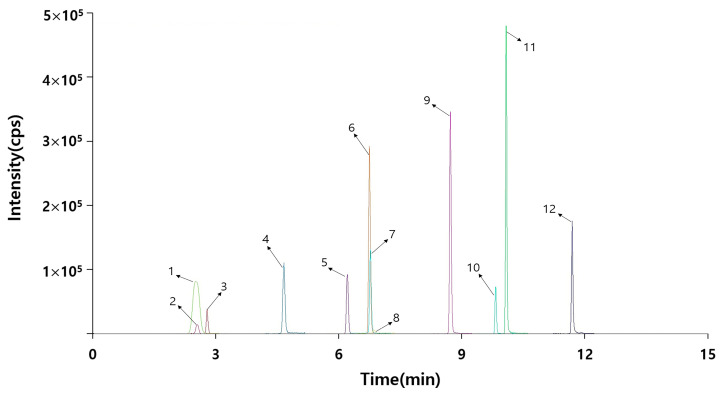
Total ion chromatogram (TIC) of 12 pesticide residues.

**Figure 2 molecules-28-05589-f002:**
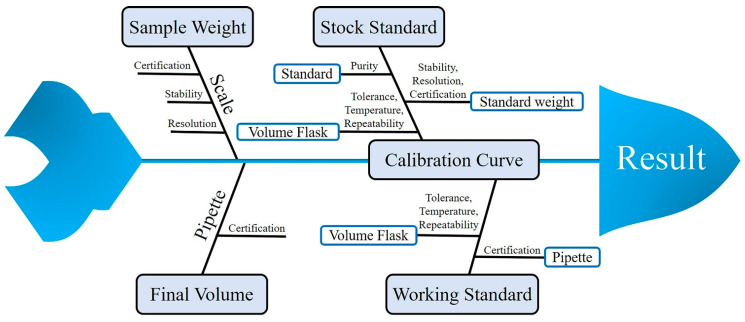
Measurement uncertainty diagram.

**Table 1 molecules-28-05589-t001:** Selected 12 pesticides and their analysis conditions.

Compound	Formula	Calculated *m*/*z*	Experimental *m*/*z*	Ionization Mode	Fragment Ion (*m*/*z*)	Mass Error (ppm)
Acequinocyl	C_24_H_32_O_4_	384.2295	407.2638	[M + NH_4_]^+^	343.2288	0.3
Boscalid	C_18_H_12_C_l2_N_2_O	342.0321	343.0399	[M + H]^+^	307.0651	2.9
Cyflumetofen	C_24_H_24_F_3_NO_4_	447.1615	465.1995	[M + NH_4_]^+^	173.0222	0.9
Dinotefuran	C_7_H_14_N_4_O_3_	202.1060	203.1138	[M + H]^+^	129.0911	−0.1
Flonicamid	C_9_H_6_F_3_N_3_O	229.0457	230.0535	[M + H]^+^	203.0442	0.5
Fluopyram	C_16_H_11_ClF_6_N_2_O	396.0458	397.0536	[M + H]^+^	173.0222	−1.8
Procymidone	C_13_H_11_C_l2_NO_2_	283.0161	301.0505	[M + NH_4_]^+^	284.0272	3.7
Propamocarb	C_9_H_20_N_2_O_2_	188.1519	189.1597	[M + H]^+^	102.0559	−1.3
Pyridaben	C_19_H_25_ClN_2_OS	364.1370	365.1448	[M + H]^+^	309.0840	0.7
Spirodiclofen	C_21_H_24_C_l2_O_4_	410.1046	411.1124	[M + H]^+^	313.0398	0.2
Spirotetramat	C_21_H_27_NO_5_	373.1883	374.1962	[M + H]^+^	330.2078	0.1
Spirotetramat-enol	C_18_H_23_NO_3_	301.1672	302.1750	[M + H]^+^	216.1031	−0.4

**Table 2 molecules-28-05589-t002:** Validation parameters of the developed UHPLC-QTOF-MS method.

Compound	r^2^	Coefficient of Variation (%)			Recovery (%)			LOD ^a^ (µg/kg)	LOQ ^b^ (µg/kg)	ME (%) ^c^	MU (%) ^d^
		10 µg/kg	50 µg/kg	100 µg/kg	10 µg/kg	50 µg/kg	100 µg/kg				
Acequinocyl	0.99985	7.1	2.1	3.3	97.7 ± 0.7	97.8 ± 1.0	99.5 ± 3.3	3.2	9.7	−13	9.1
Boscalid	0.99992	19.1	4.5	2.7	89.2 ± 1.7	98.2 ± 2.2	92.8 ± 2.5	2.1	6.3	−7	12.1
Cyflumetofen	0.99962	5.4	1.8	1.4	97.0 ± 0.5	99.7 ± 0.9	90.1 ± 1.2	2.3	7.0	3	16.2
Dinotefuran	0.99951	16.3	7.7	5.9	92.3 ± 1.5	95.7 ± 3.7	87.7 ± 5.2	2.8	8.4	10	16.3
Flonicamid	0.99960	10.1	1.8	4.3	95.3 ± 1.0	98.9 ± 0.9	94.3 ± 4.0	2.2	6.6	−1	15.2
Fluopyram	0.99958	19.0	3.7	1.8	94.7 ± 1.8	97.2 ± 1.8	89.5 ± 1.6	2.4	7.3	1	14.5
Propamocarb	0.99983	17.7	4.4	3.5	95.2 ± 1.7	98.4 ± 2.2	88.7 ± 3.1	2.1	6.2	1	11.2
Procymidone	0.99970	15.7	5.5	6.4	93.7 ± 1.5	98.0 ± 2.7	97.1 ± 6.2	2.9	8.7	7	12.8
Pyridaben	0.99996	15.1	4.2	3.2	92.6 ± 1.4	98.1 ± 2.1	92.4 ± 3.0	2.3	7.0	−5	13.9
Spirodiclofen	0.99977	12.7	3.6	0.8	86.9 ± 1.1	96.1 ± 1.7	93.0 ± 0.8	2.5	7.6	−5	18.6
Spirotetramat	0.99992	19.0	4.6	2.5	88.9 ± 1.7	97.6 ± 2.3	93.5 ± 2.3	3.0	9.0	−9	13.5
Spirotetramat-enol	0.99993	19.1	6.2	2.0	87.5 ± 1.7	90.4 ± 2.8	81.7 ± 1.6	1.4	4.1	−12	11.6

^a^ LOD: limit of detection, ^b^ LOQ: limit of quantification, ^c^ ME: matrix effect, ^d^ MU: measurement uncertainty.

**Table 3 molecules-28-05589-t003:** Uncertainty of sample weight and final volume.

Parameter	Value (*x_i_*)	Source		Type	Standard Uncertainty (*u*)	Combined Standard Uncertainty (*u*_c_)	Relative Standard Uncertainty (*u*_r_)
Sample weight	10.0335	Balance	Certification	B	0.000050	0.000078	0.000008
			Readability	A	0.000029		
			Stability	A	0.000052		
Final volume	10	Pipette	Certification	B	0.006500	0.006500	0.000650

**Table 4 molecules-28-05589-t004:** Uncertainty of calibration curve.

Source		Value (*x_i_*)	Standard Uncertainty (*u*)	1st Combined Standard Uncertainty (*u*_c_)	2nd Combined Standard Uncertainty (*u*_c_)
Stock standard solution (100 mg/L)	Purity	0.999	0.000577	0.696323	
	Balance	0.01	0.000061		
	Volumetric flask	100	0.328927		
Working standard solution (1 mg/L)	Stock standard solution	100	0.696323	0.007717	
	Pipette	1	0.000500		
	Volumetric flask	100	0.328927		
Calibration curve concentration	Acequinocyl	9.77	0.433570	0.433570	0.445305
	Boscalid	8.92	0.530122	0.530122	0.538169
	Cyflumetofen	9.70	0.776454	0.776454	0.782973
	Dinotefuran	9.23	0.746118	0.746118	0.752261
	Flonicamid	9.53	0.710811	0.710811	0.717680
	Fluopyram	9.47	0.680736	0.680736	0.687816
	Propamocarb	9.52	0.519986	0.519986	0.529318
	Procymidone	9.37	0.591857	0.591857	0.599817
	Pyridaben	9.26	0.631509	0.631509	0.638802
	Spirodiclofen	8.69	0.799879	0.799879	0.804963
	Spirotetramat	8.89	0.590417	0.590417	0.597605
	Spirotetramat-enol	8.75	0.495735	0.495735	0.504009

**Table 5 molecules-28-05589-t005:** Measurement uncertainties of selected 12 pesticide.

Compounds	Uncertainty Factor	Standard Uncertainty (*u*)	Relative Standard Uncertainty (*u*_r_)	Relative Combined Standard Uncertainty (*u*_rc_)	Combined Standard Uncertainty (*u*_c_)	Extended Uncertainty (*U*)	Measurement Uncertainty (Confidence Level about 95%, *k* = 2)
Acequinocyl	Sample weight	0.000078 g	0.000008	0.045583	0.444 μg/L	0.888 μg/L	9.74 ± 0.89 μg/L (9.1%)
	Final volume	0.006500 mL	0.000650				
	Calibration curve	0.445305 μg/L	0.045579				
Boscalid	Sample weight	0.000078 g	0.000008	0.060336	0.536 μg/L	1.073 μg/L	8.89 ± 1.08 μg/L (12.1%)
	Final volume	0.006500 mL	0.000650				
	Calibration curve	0.538169 μg/L	0.060333				
Cyflumetofen	Sample weight	0.000078 g	0.000008	0.080721	0.780 μg/L	1.561 μg/L	9.67 ± 1.57 μg/L (16.2%)
	Final volume	0.006500 mL	0.000650				
	Calibration curve	0.782973 μg/L	0.080719				
Dinotefuran	Sample weight	0.000078 g	0.000008	0.081504	0.750 μg/L	1.500 μg/L	9.20 ± 1.50 μg/L (16.3%)
	Final volume	0.006500 mL	0.000650				
	Calibration curve	0.752261 μg/L	0.081502				
Flonicamid	Sample weight	0.000078 g	0.000008	0.075310	0.715 μg/L	1.431 μg/L	9.50 ± 1.44 μg/L (15.2%)
	Final volume	0.006500 mL	0.000650				
	Calibration curve	0.717680 μg/L	0.075307				
Fluopyram	Sample weight	0.000078 g	0.000008	0.072634	0.686 μg/L	1.371 μg/L	9.54 ± 1.38 μg/L (14.5%)
	Final volume	0.006500 mL	0.000650				
	Calibration curve	0.687816 μg/L	0.072631				
Propamocarb	Sample weight	0.000078 g	0.000008	0.055604	0.528 μg/L	1.055 μg/L	9.49 ± 1.06 μg/L (11.2%)
	Final volume	0.006500 mL	0.000650				
	Calibration curve	0.529318 μg/L	0.055601				
Procymidnoe	Sample weight	0.000078 g	0.000008	0.064018	0.598 μg/L	1.196 μg/L	9.34 ± 1.20 μg/L (12.8%)
	Final volume	0.006500 mL	0.000650				
	Calibration curve	0.599817 μg/L	0.064015				
Pyridaben	Sample weight	0.000078 g	0.000008	0.068988	0.637 μg/L	1.273 μg/L	9.23 ± 1.28 μg/L (13.9%)
	Final volume	0.006500 mL	0.000650				
	Calibration curve	0.638802 μg/L	0.068985				
Spirodiclofen	Sample weight	0.000078 g	0.000008	0.092633	0.802 μg/L	1.605 μg/L	8.66 ± 1.61 μg/L (18.6%)
	Final volume	0.006500 mL	0.000650				
	Calibration curve	0.804963 μg/L	0.092631				
Spirotetramat	Sample weight	0.000078 g	0.000008	0.067225	0.596 μg/L	1.191 μg/L	8.86 ± 1.20 μg/L (13.5%)
	Final volume	0.006500 mL	0.000650				
	Calibration curve	0.597605 μg/L	0.067222				
Spirotetramat-enol	Sample weight	0.000078 g	0.000008	0.057605	0.502 μg/L	1.005 μg/L	8.72 ± 1.01 μg/L (11.6%)
	Final volume	0.006500 mL	0.000650				
	Calibration curve	0.504009 μg/L	0.057601				

**Table 6 molecules-28-05589-t006:** Concentration ranges of 12 pesticide residues analyzed in sweet peppers collected from 15 areas in Gyeongnam in Korea.

	Pesticide Detection Concentration Range (mg/kg)											
	Acequinocyl	Boscalid	Cyflumetofen	Dinotefuran	Flonicamid	Fluopyram	Procymidone	Propamocarb	Pyridaben	Spirodiclofen	Spirotetramat	Spirotetramat-Enol
Gangseo	-	-	-	0.012	-	-	-	-	-	-	-	-
Geoje	-	0.018–0.623	-	-	-	-	-	-	0.01	-	-	-
Geochang	0.047	-	-	-	0.056	-	-	-	-	-	-	-
Goseong	-	0.013–1.316	0.036–0.091	0.032–0.622	0.02–0.109	0.018–0.162	-	0.017–0.04	0.013–0.37	-	0.125–0.585	-
Gimhae	0.115–0.796	0.131–0.728	-	0.027–0.224	0.016–0.119	-	-	-	0.035–0.102	-	0.041–0.231	-
Miryang	-	0.016–0.3	0.501	0.019	0.036–0.119	-	-	-	0.048–0.166	-	0.016	-
Sancheong	-	0.154	-	0.093–0.278	-	-	-	-	0.052	-	0.848–1.626	-
Uiryeong	-	0.404	0.033	0.036–0.12	0.015–0.02	-	-	-	0.217–0.362	-	0.493	-
Jinju	0.015–0.035	0.011–0.885	0.27	0.012–1.335	0.017–0.485	0.024–0.306	0.02–0.041	0.012–0.028	0.029–0.118	-	0.032–1.041	-
Changnyeong	0.02–0.113	0.121–0.47	-	0.071–0.723	0.129	-	-	0.059	0.024–0.568	-	0.026–0.349	-
Changwon	0.018–0.393	0.048–0.755	0.16–0.309	0.013–1.24	0.012–0.056	0.087–0.227	-	0.01–0.02	0.013–0.964	-	0.017–1.236	-
Tongyeong	-	-	-	0.07–0.376	0.017	-	-	0.019–0.041	0.15–0.515	-	-	-
Hadong	0.163	-	-	0.01–0.187	0.012	-	-	0.078	0.32	-	-	-
Haman	0.021–0.072	0.03–0.733	0.208	0.035–0.513	0.01–0.167	0.153	-	0.019–0.04	0.01–0.198	-	0.021–0.471	-
Hapcheon	-	0.018	-	-	-	-	-	-	-	-	0.121–0.272	-
	0.015–0.796	0.011–1.316	0.033–0.501	0.02–0.041	0.01–0.485	0.018–0.306	0.02–0.041	0.01–0.078	0.01–0.964	-	0.016–1.626	-

**Table 7 molecules-28-05589-t007:** Analytical conditions of UHPLC-QTOF-MS.

**Mobile Phase A**	5 mM ammonium acetate & 0.1% formic acid in water			
**Mobile Phase B**	5 mM ammonium acetate & 0.1% formic acid in methanol			
**Gradient**	Time (min)	A (%)	B (%)	Flow (mL/min)
	Initial	100	0	0.1
	0.2	100	0	0.1
	0.3	100	0	0.3
	0.5	50	50	0.3
	2.5	45	55	0.3
	5.5	25	75	0.3
	7.5	15	85	0.3
	8.3	0	100	0.3
	12.0	0	100	0.3
	12.1	100	0	0.3
	14.8	100	0	0.3
	14.9	100	0	0.1
	15.0	100	0	0.1
**Injection volume**	10 μL			
**Column temperature**	40 °C			
**Ionization mode**	Electrospray ionization mode (positive mode)			
**Source and gas parameters**	Ion source gas 1–60 psi, curtain gas—30 psi, temperature—450 °C, ion source 2–40 psi, CAD gas—7			
**QTOF, MS/MS**	TOF start mass—100 Da, declustering potential—80 V, collision energy—10 V, TOF stop mass—1000 Da, DP spread—0 V, CE spread—0 V, accumulation time—0.25 s			

## Data Availability

Not applicable.

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
