# Peer review of "Ultra-High-Performance Liquid Chromatography Quadrupole Time-of-Flight Mass Spectrometry for Simultaneous Pesticide Analysis and Method Validation in Sweet Pepper"

_molecules, 2023, doi:10.3390/molecules28145589_

Round 1
Reviewer 1 Report
Comments and Suggestions for Authors
The article titled” Multiple simultaneous analyses and method validation of pesticide residues in sweet pepper (Capsicum annuum) using ultra-3 high-performance liquid chromatography /electrospray ioniza-4 tion quadrupole time-of-flight mass spectrometry “ have been reviewed. The authors developed a simultaneous multicomponent analytical method based on UHPLC-QTOF-MS for the estimation of residual pesticide levels.
The study is of analytical significance and can be accepted after incorporating the following suggestions:
a) The title is too much long that may be revised
b) The abstract needs to be revised by adding more information about the analytical findings. Moreover, the abbreviation (UHPLC-QTOF) used for “ultraperformance liquid chromatography/electrospray ionization quadrupole time-of-flight mass spectrometry should also be rechecked and corrected throughout the manuscript by adding abbreviated letters for “mass spectrometry” as well. e.g., UHPLC-QTOF-MS or UHPLC-QTOF-MS/MS accordingly.
c) The introduction may be further improved by adding latest relevant background information supported with latest references (2022 &2023).
d) Method validation and analysis results are well presented however, discussion may be improved by adding more comparative description, how the presented analytical method is better than the other methods being used for the similar analytical characterization?
e) Some information about the fragmentation patterns of studied pesticides may also be included.
f) The manuscript once should be thoroughly rechecked for any grammatical mistake
Comments on the Quality of English LanguageModerate editing of English language is required . The manuscript should also be thoroughly recheked for any grammatical mistake.
Reviewer 2 Report
Comments and Suggestions for Authors
The paper by Bang et al. is interesting and reports a straight-forward method for the analysis of pesticides in sweet peppers by UPLC-ESI-QTOF. The method validation seems to be carefully done and the method reported suitable to evaluate the safety of sweet peppers to consumers.
However, I think that the paper needs to be revised in order to make it more usable and readable for other researchers. Firstly, I would like to know more about the selection of 12 pesticides. You mentioned that it was made based on high detection frequency. However, only 10 of these were detected in sweet peppers. Please, clarify this and show the supporting literature references. How characteristic are the fragments ions shown in Table 1? It would be nice to have a figure with the selected pesticide structures and their characteristic fragmentation shown.
Secondly, the actual method used is unclearly presented. For example, ESI is mentioned in the title but not in the experimental part. The ion mode is not mentioned either but the data shows it to be positive ionization. The detailed experimental MS conditions and parameters should be added to the Section 3.2. and in addition, the MS method used should be clarified. You refer to MRM in lines 84-86, but typically MRM is performed by triple quadrupole instruments as TOF is not capable to measure fixed m/z values. How do you perform MRM by QTOF? By using some kind of deconvolution tool? I agree that QTOF is an excellent tool to these studies due to its high resolving power. However, I must note that all QTOF instrumentations are not yet suitable for quantitative work, even though nowadays, the manufacturers are aiming to produce quantitative QTOFs.
What are the 140 pesticides in table S1? Commercial standards? Please, clarify these. Show the data with four decimals and also the mass error for each compound to justify the identification and to show the accuracy of the QTOF analyses. Please, explain the declustering potential as it is a specific parameter for AB Sciex instruments. For me, these values seem to be rather high when used in ion source (up to 150 eV) in comparison to the typical values used in EI (70 eV). How can you be sure that there is no in-source fragmentation happening?
Thirdly, I would like to have more discussion in Section 2.4., which is showing the applications of the developed method. I think that this is the main aim of the study. What is the applicability of the method and what are the real-life applications? There is a lot of data in Table 7 regarding different sweet pepper varieties providing a basis for deeper discussions. In addition, I’m a little bit worried about the matrix effects (see also the minor comment). Could it be possible that the method underestimates the pesticide contents in real biological samples?
Minor comments and corrections:
- Keywords are typically presented in alphabetical order.
- Line 41: oral intake is more dangerous.
- Lines 64-67 and 76-79: Please, note that QTOF is also an MS/MS technique. I assume that you referring to triple quadrupole and MRM techniques by these sentences.
- Line 81: experiment
- Lines 84-86: As mentioned above, this part is confusing. Please, clarify MRM and how to perform MRM by QTOF.
- m/z always in italics.
- Table 2: the numbers of elements in subscripts, the charges of the ion in superscripts, m/z in italics
- Lines 111-112: You suggest that external calibration using a standard can be used for quantitative purposes. Is this true? Can you evaluate it only by using standards solution? How about matrix effects?
- Lines 118-119: to show sensitivity? Please, clarify.
- Figure 1. I suppose that here should be extracted ion chromatograms. Please, show the ions and the +/- m/z range used for EICs.
- Lines 143-144: various factors were taken into account or tested?
- Figure 2. I would use colors in the figure and also adjust the font size. The figure is very informative but hard to read as such. The same is true for Tables 4-6. Could they be somehow redesigned? Do they need to be in the actual manuscript or could they be moved to supplementary material?
- Please, make Table 7 visually more readable. Please, clarify n, a and b. Are a and b correctly used in the table? See for example, Haman (77/64): do this mean that 77 Haman sweet peppers were analyzed and 64 of these samples contained pesticides? What is then (4) for acequinocyl? 4 cannot be b = “number of detected pesticides” as we are talking about one pesticide.
- Line 183: Please, explain the abbreviation MRL.
